# Effect of the Strain Rate and Fiber Direction on the Dynamic Mechanical Properties of Beech Wood

**Shumeng Pang, Yingjing Liang \*, Weijun Tao, Yijie Liu, Shi Huan and Hongfa Qin**

School of Civil Engineering, Guangzhou University, 230 Wai Huan Xi Road, Guangzhou 475300, China; pangshumeng@e.gzhu.edu.cn (S.P.); mindconcept@163.com (W.T.); liuyijie1987@outlook.com (Y.L.); guangzhoudaxue2014@163.com (S.H.); kiteeee@163.com (H.Q.)

\* Correspondence: yjliang@gzhu.edu.cn; Tel.: +86-136-6076-7801

**Abstract:** As a macroscopically orthotropic material, beech wood has different mechanical properties along the fiber direction and the direction perpendicular to the fiber direction, presenting a complicated strain rate sensitivity under impact or blast loadings. To understand the effect of the strain rate on the mechanical properties of beech wood, dynamic compression tests were conducted for the strain rate range of 800 s$^{-1}$–2000 s$^{-1}$, and quasi-static compression tests for obtaining the static mechanical properties of beech wood were also performed for comparison. The fiber direction effect on the mechanical properties was also analyzed, considering two loading directions: one perpendicular to the beech fiber direction and the other parallel to the beech fiber direction. The results show that beech wood for both loading directions has a significant strain rate sensitivity, and the mechanical properties of beech wood along the fiber direction are superior to those along the direction perpendicular to the fiber direction. An analysis of the macrostructures and microstructures of beech specimens is also presented to illustrate the failure mechanisms. The beech wood, as a natural protective material, has special dynamic mechanical properties in the aspect of transverse isotropy. This research provides a theoretical basis for application in protective structures.

**Keywords:** beech wood; mechanical properties; strain rate sensitivity; fiber direction; failure mechanisms

---

## 1. Introduction

Wood, as a material well-known for its renewable nature, has been applied to construction and buildings since ancient times. In particular, beech wood has been used for several applications in the restoration and construction of traditional Chinese architecture, including palaces, temples, and classical gardens. Beech wood always plays an irreplaceable role in Chinese architecture [1,2]. Beech wood and wood composites have been widely used in construction and buildings in many European countries in recent decades, due to the fact that they are renewable, efficient, and available in a variety of shapes [3,4]. Additionally, wood, similarly to other materials, such as honeycomb material [5], foam material [6–8], and sandwich composite material [9], has also been applied in sandwich structures as an energy-absorbing material or protective material [10,11], and has the advantages of being cheaper and displaying a better energy-absorbing ability, while still having a higher strength after severe deformation.

With the wide application of wood in construction and building, the safety of wooden structures is a hot research issue in disaster prevention and mitigation research [12,13]. In particular, for the joint between wood members, there are many types of joints, and their mechanical responses under static loadings are different [14]. The mechanical responses should be analyzed according to the static mechanical properties of wood materials.

Numerous studies on the static mechanical properties of wood, such as the compression and tensile strengths, have shown that the static mechanical properties of different types of wood are different, and the yield stress and the elastic modulus of wood along the wood fibers are higher than those of wood along the direction perpendicular to the wood fibers. Furthermore, it has been demonstrated that the mechanical properties of wood and the deformation process at a high strain are significantly affected by the dimensions and shape of wood cells [15–17], temperature and moisture content [18,19], and the fiber direction [20]. However, when wood structures are subjected to blast or impact loads, the mechanical characteristics of structural members, especially those of the joints, are very complex, which is different from the static mechanical characteristics of structural members. Moreover, the wood sandwiched between steel plates, as an energy-absorbing material, presents a dynamic non-linear mechanical response when it is subjected to blast or impact loads [10,11]. In order to analyze the safety of a structure under dynamic loads, it is necessary to perform accurate testing to understand the dynamic mechanical properties of wood materials.

The split Hopkinson pressure bar (SHPB) [21], as a common experimental device for testing the dynamic mechanical properties of materials, can solve the coupled problem between the stress wave effect and strain rate effect of materials under high strain rates. The SHPB device can produce different strain rates of test specimens by controlling the impact velocities of the striker bar, and has been applied in testing for dynamic mechanical properties of different kinds of materials under different high strain rates, such as foam materials [5–8], concrete materials [22,23], metal materials [24,25], and composite materials [26,27].

For studying the dynamic mechanical properties of wood under SHPB, Reid et al. [28] applied the SHPB technique to analyze the dynamic mechanical compression behavior of woods along the different fiber directions. Renaud et al. [29] evaluated the dynamic mechanical properties of three species of hardwoods in three different directions by SHPB under different strain rates, and also considered the effect of different swelling liquids on the mechanical behavior. They found that the liquid inside the specimens could not flow out of the pores, and the specimens presented a stiff material at high strain rates. Widehammar [30] investigated the influence of the strain rate, moisture content, and loading direction on the mechanical properties of spruce wood, and found that the spruce wood with full saturation exhibited more strain rate sensitivity than that with fiber saturation. Tagarielli et al. [31] studied the effect of the strain rate on the mechanical properties of polyvinyl chloride (PVC) foams and balsa wood by low strain rate tests, intermediate strain rate tests, and high strain rate compression tests, and analyzed their different strain rate sensitivities. Moilanen et al. [32] studied the radial mechanical properties of Norway spruce wood under high strain rates by SHPB, and analyzed the effect of room temperature and an elevated temperature on the test result, in addition to developing a simple wood compression model for the mechanical pulping process of wood. Hu et al. [26] studied the effect of the strain rate on the mechanical behavior of bamboo by employing static compression tests and SHPB compression tests, and also analyzed the effect of bamboo fibers with different directions on the test results of bamboo. They indicated that the mechanical properties of bamboo with different fiber directions are significantly different and all exhibited strain rate sensitivity. Wouts et al. [27,33] studied the effect of the strain rate on the mechanical properties of spruce wood and beech wood, and the tests were conducted by three experimental pieces of apparatus with and without rigid lateral confinement to cover different strain rates from $0.001 \text{ s}^{-1}$ to $600 \text{ s}^{-1}$. They also discussed the energy absorption of wood along three orthotropic directions under different strain rates and investigated the influence of the strain rate on the initial crushing stress and plateau stress.

As shown above, research on wood has focused on the influence of moisture content, density, fiber direction, strain rate, and temperature on the mechanical behavior of wood materials. The results show that both the strain rate and the fiber direction influence the mechanical properties of woods, especially the dynamic mechanics. The effect of fiber direction along the longitudinal direction is more significant than that along both the radial direction and the tangential direction. However, there are a few references that have focused on the dynamic properties of beech wood. Wouts et al. [27,33] studied

the effect of the strain rate on the mechanical properties of both hardwood and softwood materials. However, the maximum strain rate they employed was 600 s$^{-1}$, which was considered a low strain rate—it did not reach the magnitude of a high strain rate of $10^3$ s$^{-1}$.

This work studies the dynamic mechanical properties of beech wood under high strain rates, and investigates the effect of both strain rates and fiber direction on the mechanical properties of beech wood. The dynamic mechanical properties are obtained from SHPB tests, and the static mechanical properties are obtained from static tests for comparison. Further, the deformation progress of beech specimens in static tests is acquired through continuous photographic techniques using the Nikon camera 5500, and the failure mechanisms of beech specimens in SHPB tests are analyzed through pictures from a photo camera and scanning electron microscope (SEM).

## 2. Materials and Methods

In this study, the test materials were extracted from the middle part of a tree trunk, which was naturally grown beech wood imported from Germany by SHUN FENG FA WOOD Company, located in Huizhou City, Guangdong Province, China. Beech wood is characterized as a macroscopically orthotropic material rather than an isotropic material. For comparison, the whole test specimens were taken from the same position along the longitudinal direction of the beech tree to reduce the difference among specimens. They were machined in radial and longitudinal directions by machining, as shown in Figure 1. The moisture content and density of the beech wood were about 10% and 0.73 g/cm$^3$ at room temperature, respectively.

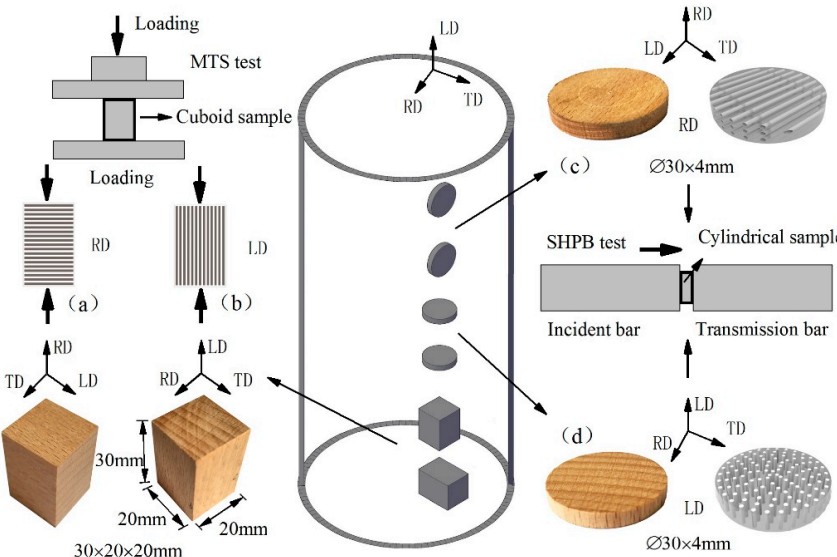

**Figure 1.** Real specimen pictures and a schematic diagram of test specimens for (**a**) the Material Testing Simulation (MTS) test in the radial direction (RD), (**b**) MTS test in the longitudinal direction (LD), (**c**) split Hopkinson pressure bar (SHPB) test in the RD, and (**d**) SHPB test in the LD.

For quasi-static compression tests, the beech specimens were regular cuboid-shaped samples with dimensions of $30 \times 20 \times 20$ mm, as shown in Figure 1a, corresponding to the radial direction (RD), longitudinal direction (LD), and tangential direction (TD) of beech specimens loaded in the radial direction, respectively. The samples' dimensions of $30 \times 20 \times 20$ mm, as shown in Figure 1b, correspond to the longitudinal direction, tangential direction, and radial direction of the beech specimen loaded in the longitudinal direction, respectively. The results from some references show that the mechanical properties of wood materials along the longitudinal direction are significantly different from those along both the radial direction and tangential direction, and the compressive strength of wood materials along both the longitudinal direction and radial direction is larger than that along the tangential direction. Therefore, in this study, according to the fiber direction of beech wood, only the effects

of the longitudinal direction and radial direction on the mechanical properties of beech wood were considered. Therefore, two groups of beech specimens were used for quasi-static compression tests: one group was beech specimens with a direction perpendicular to wood fibers along the RD, as shown in Figure 1a, while the other one was those with wood fibers along the LD, as shown in Figure 1b. Six samples used for quasi-static compression tests were prepared for each group, and a total of twelve samples were prepared and tested.

As for the SHPB compression tests, the beech specimens were machined into a cylinder shape with a dimension of 30 mm in diameter and 4 mm in length. As with the beech specimens used for quasi-static compression tests, according to the fiber direction of beech wood, two groups of beech specimens were also employed for SHPB compression tests: one group was beech specimens with a direction perpendicular to wood fibers along the RD, as shown in Figure 1c, while the other one was those with wood fibers along the LD, as shown in Figure 1d. Three samples used for SHPB compression tests were prepared for a strain rate of each group, and a total of twenty-four samples were prepared and tested.

### 2.1. Quasi-Static Compression Tests

Quasi-static compression tests were performed to acquire the quasi-static mechanical properties of beech wood by a standard servo-hydraulic system Material Testing Simulation (MTS), which had a loading capacity of 600 kN. The tests were conducted under displacement control mode with the actuator speed of 6 mm/min, corresponding to an estimated strain rate of $3.33 \times 10^{-3}$ s$^{-1}$. The beech specimens were sandwiched between the moving platen and stationary platen, and the two contact interfaces were evenly smeared with Vaseline lubricant to reduce the interfacial friction effect. Two loading directions, with one being the LD parallel to the loading direction and the other one being the LD perpendicular to the loading direction, were adopted to load the specimens, as shown in Figure 1. A data acquisition system of MTS was used to automatically record the load-displacement histories. Additionally, the continuous photographic technique from the Nikon camera 5500 was used to automatically record the deformation histories of beech specimens under quasi-static loading.

### 2.2. Dynamic Compression Tests

Dynamic compression tests were conducted to acquire the dynamic mechanical properties of beech wood under different impact strain rates by SHPB apparatus, as shown in Figure 2. Typical SHPB apparatus mainly consists of a launch system, a striker bar, an incident bar, a transmission bar, and a data acquisition system, etc., and the data acquisition system included a high dynamic strain indicator (KD6009) and a DPO2000 oscilloscope. Both the incident and transmission bars had the same dimension of Ø40-2000 mm, the striker bar was Ø50-400 mm, and the absorption bar was Ø40-400 mm. These bars all belonged to the same ultra-high strength stainless steel. The strain rate ranged from $10^2$ s$^{-1}$ to $10^4$ s$^{-1}$.

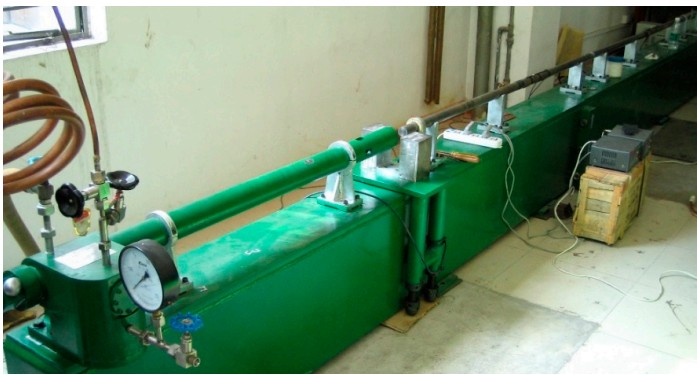

**Figure 2.** *Cont.*

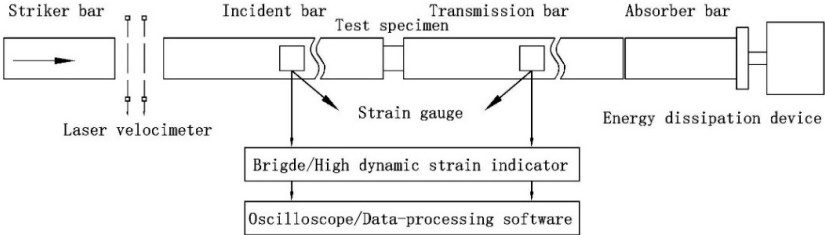

**Figure 2.** Photograph and schematic diagram of the split Hopkinson pressure bar (SHPB) test apparatus.

As shown in Figure 2, the beech specimens for SHPB tests were sandwiched between the incident bar and transmission bar. A thin layer of Vaseline lubricant was spread on the two interfaces between the beech specimen and both the incident bar and transmission bar to reduce the interfacial friction effect. When the launch system controls the gas pressure to launch the striker bar to impact the incident bar, an incident stress wave propagating from the incident bar to the beech specimen is generated. Due to the impedance mismatch between the beech specimen and pressure bars, the incident stress wave will reflect and transmit at the interface between the beech specimen and the incident bar. Part of the stress wave will be reflected back to the incident bar, while the remainder will transmit through the beech specimen, and will then reflect and transmit again at the interface between the beech specimen and the transmission bar, while some of the remaining stress wave will be transmitted through the transmission bar. The stress wave signals were measured by semiconductor strain gauges on the pressure bar, the incident wave signal $\varepsilon_I(t)$ and reflected wave signal $\varepsilon_R(t)$ were obtained from the semiconductor strain gauge on the incident bar, and the transmitted wave signal $\varepsilon_T(t)$ was obtained from that on the transmission bar. The strain rate $\dot{\varepsilon}(t)$, the dynamic strain $\varepsilon(t)$, and the dynamic stress $\sigma(t)$ of the beech specimen during SHPB tests could be deduced based on the one-dimensional stress wave propagation theory—the equations for which are as follows [21]:

$$\dot{\varepsilon}(t) = \frac{C_0}{L_s}(\varepsilon_I(t) - \varepsilon_R(t) - \varepsilon_T(t)) = -\frac{2C_0}{L_s}\varepsilon_R(t) \tag{1}$$

$$\varepsilon(t) = \int_0^t \dot{\varepsilon}(t)dt = -\frac{2C_0}{L_s}\int_0^t \varepsilon_R(t)dt \tag{2}$$

$$\sigma(t) = \frac{EA_0}{2A_s}(\varepsilon_I(t) + \varepsilon_R(t) + \varepsilon_T(t)) = \frac{EA_0}{A_s}\varepsilon_T(t) \tag{3}$$

where $C_0$, $E$, and $A_0$ are the stress wave velocity, Young's modulus, and the cross-section area of the pressure bars, respectively. $L_s$ and $A_s$ are the length and the original cross-section area of the specimen, respectively.

The equations deduced above are the engineering strain and stress of the material. Given that the material is incompressible, the true strain $\varepsilon_T(t)$ and the true stress $\sigma_T(t)$ of the material can be obtained:

$$\varepsilon_T(t) = -\ln(1 - \varepsilon(t)) \tag{4}$$

$$\sigma_T(t) = (1 - \varepsilon(t))\sigma(t) \tag{5}$$

For convenience, the quasi-static compression tests for beech specimens along the fiber direction (LD) are called QTLD, while those along the direction perpendicular to fibers (RD) are called QTRD. The dynamic compression tests for beech specimens along the fiber direction (LD) are called DTLD, while those along the direction perpendicular to fibers (RD) are called DTRD. Additionally, all the tests were conducted at a room temperature of around 25 °C.

## 3. Results and Discussion

### 3.1. Quasi-Static Compression Test Results

As shown in Figure 3, the typical stress–strain curves of the beech specimens were obtained from QTLD and QTRD, respectively. The typical stress–strain curve from QTLD at the last stage tends to decrease, the typical stress–strain curve from QTRD in the hardening stage tends to increase, and the beech specimen under QTRD produces large deformation. For comparison, the typical stress–strain curves from QTRD and QTLD are presented within a small deformation range. Figures 4 and 5 show the deformation progression of beech specimens in QTRD and QTLD at different times, respectively, and the pictures obtained using a Nikon camera 5500 show the dynamic trend of fracturing and the process of failure in real time.

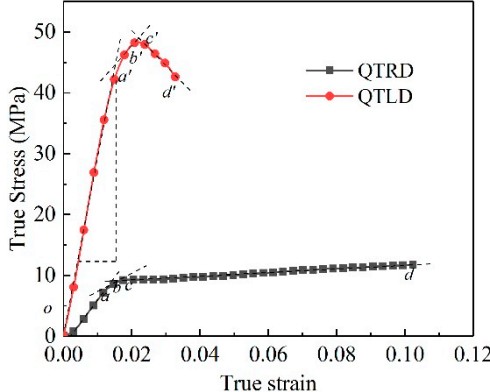

**Figure 3.** Typical quasi-static stress–strain curves of beech specimens in QTLD and QTRD.

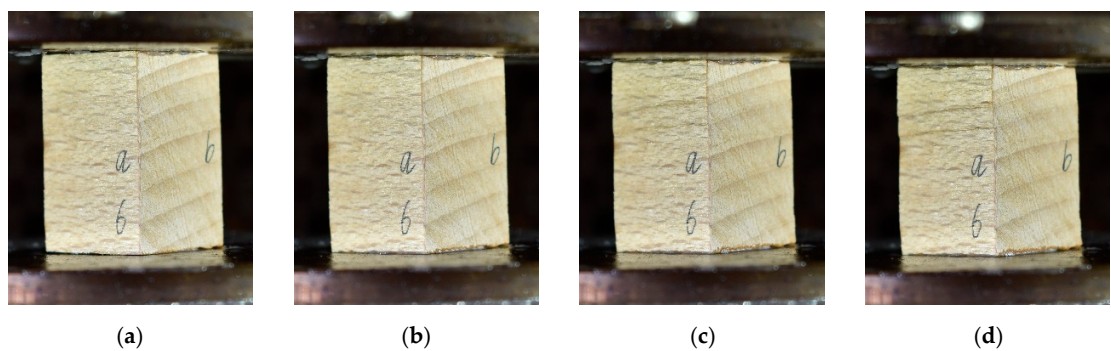

| (a) | (b) | (c) | (d) |

**Figure 4.** Deformations of beech specimen in QTRD at different times of (**a**) 0 s, (**b**) 6.5 s, (**c**) 12 s, and (**d**) 14 s.

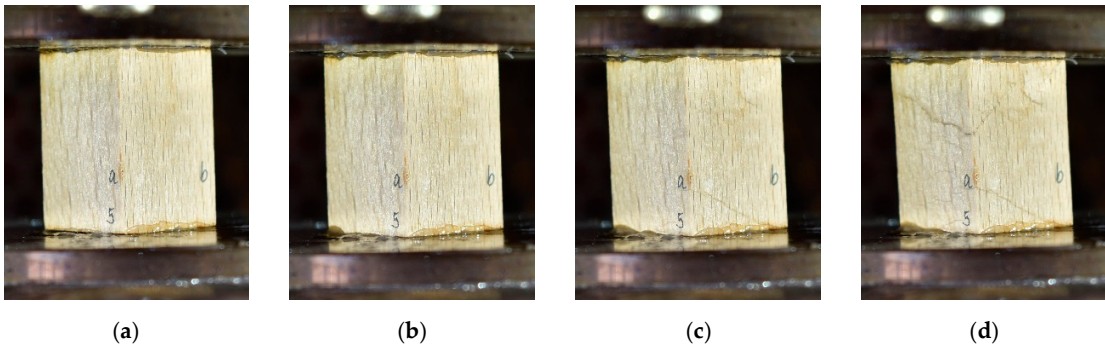

| (a) | (b) | (c) | (d) |

**Figure 5.** Deformations of beech specimen in QTLD at different times of (**a**) 0 s, (**b**) 6.5 s, (**c**) 12 s, and (**d**) 14 s.

From the results of QTLD and QTRD, it can be observed that the yield stress value of the beech specimen in QTRD is much lower than that in QTLD, and the ultimate strength of the beech specimen in QTRD is also lower than that in QTLD within a certain strain level. It can be concluded that the fiber direction of beech wood is a very important factor that influences the mechanical properties of beech specimens. Additionally, the difference between the shapes of stress–strain curves of QTLD and QTRD is significant. There are three different stages in the stress–strain curve of QTRD, namely, the elastic deformation stage (oa), plastic evaluation stage (ac), and hardening development stage (cd). In the elastic deformation stage, the fibers start to press against each other, and their fiber spacing decreases under increasing displacement. Furthermore, when the fiber spacing reaches a certain threshold, the interaction force of fibers does not increase linearly, the beech specimen exhibits plastic deformation, and its stress–strain behavior develops with a nonlinear plastic relationship. The beech specimen shows hardening deformation with the continuous displacement load. As shown in Figure 4, during the compression process, the fibers between the rings are squeezed together and dislocated. In comparison, the trend of the stress-strain curve of QTLD shows another three stages, that is, the elastic deformation stage (o'a'), plastic evolution stage (a'c'), and softening development stage (c'd'). In the elastic deformation stage, the fibers in the longitudinal direction are loaded, and the fibers deform elastically. With increasing loads, the fibers show large deformation and change to nonlinear plastic deformation. When the total value of the displacement load reaches a certain threshold, the fibers buckle, delaminate, and fracture, and reach failure. As shown in Figure 5, during the compression process, the fibers are compressed and appear buckled and fractured, and the specimen then undergoes shear-slip failure at nearly forty-five degrees. From Figure 3, it can also be observed that the elastic stage of the beech specimen in QTLD increases more rapidly than that in QTRD, which means that the elastic modulus of beech wood along the longitudinal direction of fibers is higher than that along the direction perpendicular to fibers, which is a phenomenon caused by the fiber arrangement of beech wood. The beech wood can be a macroscopically orthotropic material, which has a fiber-reinforced character.

### 3.2. SHPB Compressive Test Results

For comparison, the DTRD and DTLD were conducted under the same impact strain rates, and here, four impact strain rates are given: 800, 1200, 1800, and 2000 s$^{-1}$. Based on Eq. (1)–(5), the strain rate, the dynamic strain, and the dynamic stress of beech specimens were obtained from DTRD and DTLD. In order to verify the validity and repeatability of the SHPB tests, some information is provided. Figure 6a presents the original wave signals, which include the incident wave signal, the reflected wave signal, and the transmitted wave signal. In the process of data calculation, these three wave signals need to be extracted within the same time duration. Figure 6b gives their relationship, showing that these three wave signals meet the one-dimensional stress wave theory, which can verify the validity of the SHPB tests.

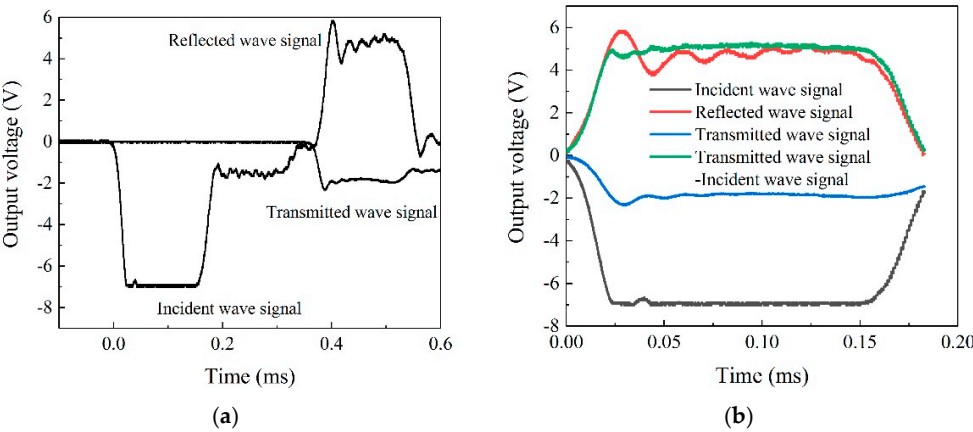

**Figure 6.** (**a**) The original wave signals from the SHPB test, and (**b**) their relationship.

Figure 7 presents some dynamic mechanical information on beech specimens from DTRD and DTLD under the same strain rate of 2000 s$^{-1}$, including strain rate versus time (Figure 7a), strain versus time (Figure 7b), stress versus time (Figure 7c), and stress versus strain (Figure 7d). Figure 7a shows that the trend of the strain rate–time curve from DTLD is similar to that from DTRD, and they all have a wider plateau of the strain rate at about 2000 s$^{-1}$. Additionally, as seen from Figure 7b, the strain–time curve from DTLD almost coincides with that from DTRD, which illustrates that they have the same strain rate.

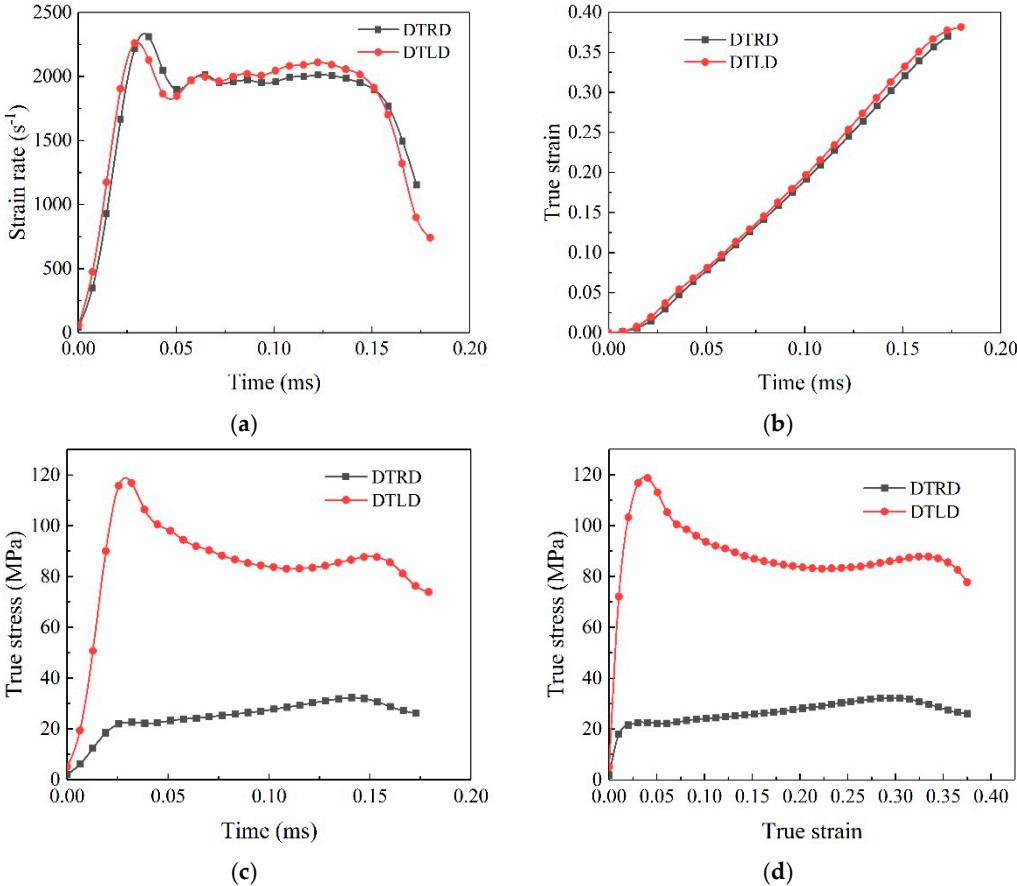

**Figure 7.** Representative curves of specimens in DTRD and DTLD under the strain rate of 2000 s$^{-1}$: (**a**) strain rate–time curves, (**b**) strain–time curves, (**c**) stress–time curves, and (**d**) stress–strain curves.

In Figure 7d, the dynamic stress–strain curve of the beech specimen in DTRD presents the same developing stages as that in QTRD within a certain strain level. However, compared with the stress–strain curve from DTRD in Figure 7d, the beech specimen in DTLD experiences the same elastic deformation stage initially, and then exhibits non-linear plastic deformation due to fiber buckling. When the fibers are loaded to reach the ultimate stress, they are in a state of instability, and the stress begins to decrease. However, with the increase in strain, the stress maintains an approximatively stable value, and nonlinear hardening deformation to the ultimate stress then occurs and begins to decrease, which is caused by the unloading effect of the incident stress wave.

### 3.3. Strain Rate Sensitivity

As shown in Figures 8 and 9, the typical stress–strain curves of beech specimens at different strain rates were obtained from DTLD and DTRD, respectively. The typical stress–strain curves of beech specimens obtained from QTLD and QTRD are also plotted for comparison, to illustrate the effect of the strain rate on the dynamic mechanical behavior of beech specimens. As seen in Figures 8 and 9, the values of yield stress and ultimate stress of beech specimens from DTLD and DTRD improve with

the increase in the impact strain rates, which shows that both beech specimens of DTLD and DTRD have obvious strain rate sensitivity.

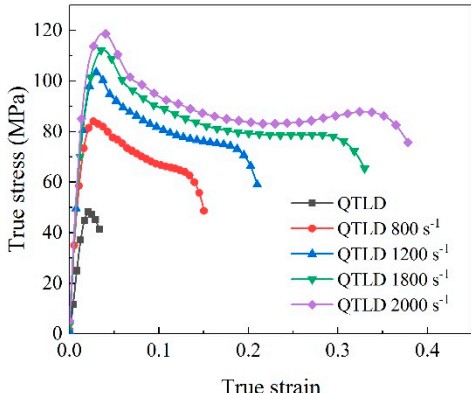

**Figure 8.** Typical stress–strain curves of beech specimens at different rates in DTLD.

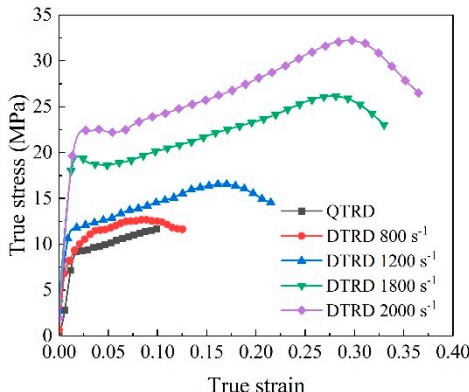

**Figure 9.** Typical stress–strain curves of beech specimens at different rates in DTRD.

Considering several parameters, including the strain rate, yield stress, and ultimate stress, Table 1 summarizes the results from quasi-static compressive tests and SHPB compressive tests. It can be observed that the values of yield stress and ultimate stress of beech specimens from DTLD and DTRD are directly proportional to the strain rate. In DTLD, when the strain rate is 2000 s$^{-1}$, the value of yield stress is 114.1MPa, which is about 1.455-, 1.167-, and 1.06-times higher than that under the strain rate of 800 s$^{-1}$, 1200 s$^{-1}$, and 1800 s$^{-1}$, respectively. Additionally, in DTRD, the value of yield stress under a strain rate of 2000 s$^{-1}$ is 20.6 MPa, which is about 2.418-, 2-, and 1.177-times higher than that under the strain rate of 800 s$^{-1}$, 1200 s$^{-1}$, and 1800 s$^{-1}$, respectively. For clear analysis, the results of the SHPB compression tests have been fitted to show the development of yield stress under different strain rates in DTLD and DTRD. The equations are as follows:

DTLD:
$$\sigma_y = 44.51339 + 0.0528\dot{\varepsilon} - 9.22597 \times 10^{-6}\dot{\varepsilon}^2, \ r^2 = 0.98617 \tag{6}$$

DTRD:
$$\sigma_y = 8.18614 - 0.00414\dot{\varepsilon} + 5.16117 \times 10^{-6}\dot{\varepsilon}^2, \ r^2 = 0.99615 \tag{7}$$

where $\sigma_y$, $\dot{\varepsilon}$, and $r^2$ are the yield stress, strain rate, and residuals of the least squares method, respectively.

**Table 1.** Results from quasi-static compressive tests and SHPB compressive tests.

| Test Condition | Loading Strain Rate (s$^{-1}$) | Yield Stress (MPa) | Ultimate Stress (MPa) |
|---|---|---|---|
| QTLD | $3.33 \times 10^{-3}$ | 44.9 | - |
| DTLD | 800 | 78.4 | 83.94 |
| | 1200 | 97.8 | 103.5 |
| | 1800 | 107.6 | 112.4 |
| | 2000 | 114.1 | 119 |
| QTRD | $3.33 \times 10^{-3}$ | 8.12 | - |
| DTRD | 800 | 8.52 | 12.7 |
| | 1200 | 10.3 | 16.6 |
| | 1800 | 17.5 | 26.17 |
| | 2000 | 20.6 | 32.26 |

Figure 10 shows the energy absorption per unit volume (EAV)–strain curves at different strain rates of specimens in DTLD and DTRD, which were acquired by integrating the stress–strain curves of Figures 8 and 9, respectively. It can be observed that the energy absorption per unit volume of the beech specimen along the fiber direction is better than that along the direction perpendicular to fibers, and the energy absorption per unit volume of beech wood is strain rate sensitive.

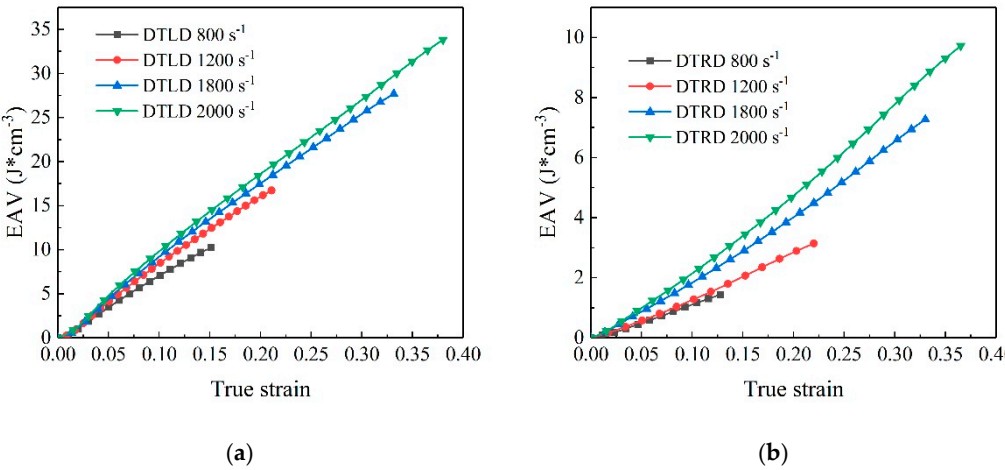

(a) (b)

**Figure 10.** Absorbed energy per unit volume under different strain rates: (**a**) DTLD results and (**b**) DTRD results.

### 3.4. Failure Mechanism

The beech specimens under high strain rate loading will produce large deformation and even be crushed. The failure mechanisms of beech specimens along the fiber direction and the direction perpendicular to fibers can be investigated at macro and micro scales. Figures 11 and 12 present the final macroscopic failure modes of beech specimens from DTLD and DTRD under different strain rates of 800, 1200, 1800, and 2000 s$^{-1}$, respectively. Figure 13 shows the microscopic failure modes of beech specimens from DTLD and DTRD under the same strain rate of 2000 s$^{-1}$, respectively. The microscopic pictures of these specimens were all obtained by SEM. The original microstructure of the beech specimen is also given for comparison. Figure 14 presents the diagram of failure modes of beech specimens under a high strain rate, and the failure modes of beech specimens under static loadings are shown for comparison. Figure 14a,b correspond to Figures 4 and 5, respectively, and Figure 14c,d correspond to Figures 11 and 12, respectively.

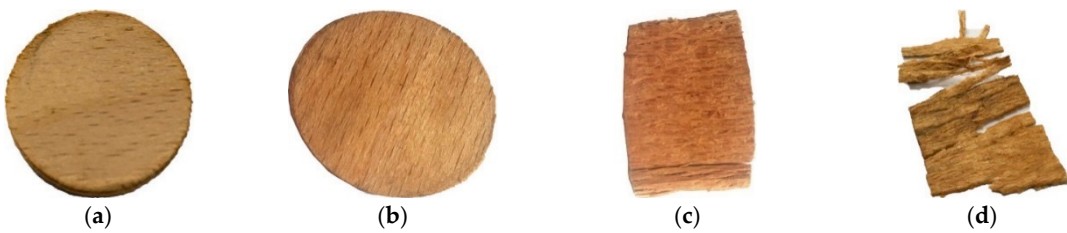

**Figure 11.** Final failure shapes of DTRD under different strain rates of (**a**) 800 s$^{-1}$, (**b**) 1200 s$^{-1}$, (**c**) 1800 s$^{-1}$, and (**d**) 2000 s$^{-1}$.

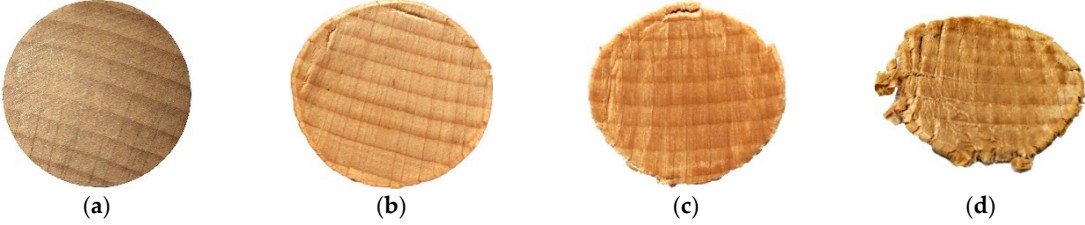

**Figure 12.** Final failure shapes of DTLD under different strain rates of (**a**) 800 s$^{-1}$, (**b**) 1200 s$^{-1}$, (**c**) 1800 s$^{-1}$, and (**d**) 2000 s$^{-1}$.

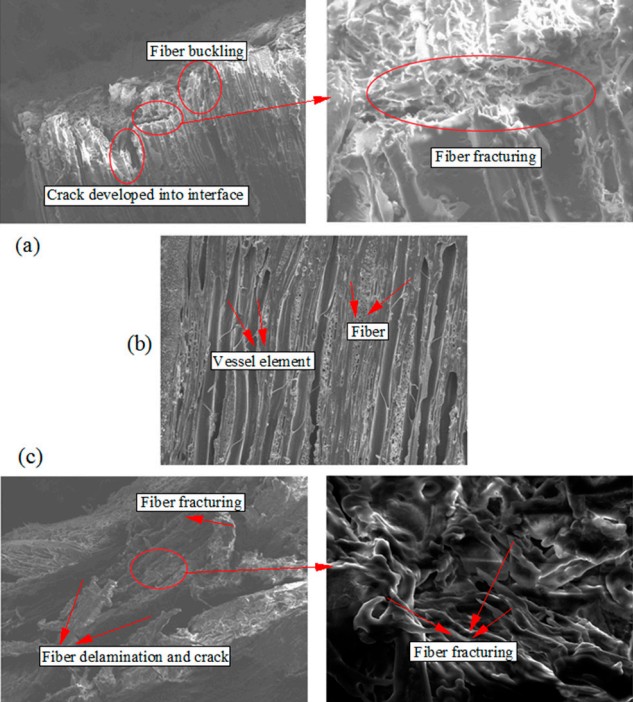

**Figure 13.** Microstructures of beech specimens (**a**) in DTLD, (**b**) before being loaded, and (**c**) in DTRD.

Figure 14 schematically shows four different failure modes of beech specimens alongside the fiber direction and direction perpendicular to the fibers under static and dynamic loadings. As shown in Figure 14a, during the compression process, the fibers between the rings are squeezed together and dislocated, and the dislocation deformation leads to the failure of beech specimens. Figure 14b shows another failure mode, where the fibers appear to be buckling and fracturing, there happens to be shear-slip failure at nearly forty-five degrees, and the normal direction of the shear-slip plane is perpendicular to the radial direction of the beech specimen.

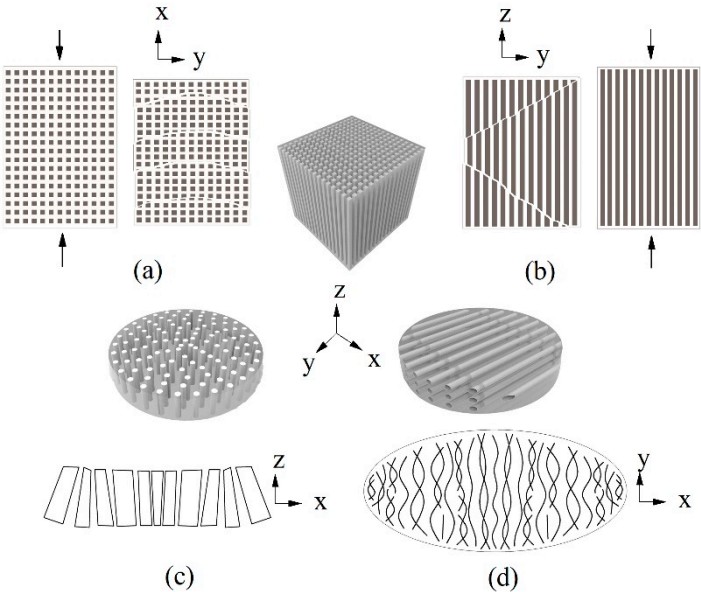

**Figure 14.** The diagram of failure modes of beech specimens under static and dynamic tests: (**a**) QTRD, (**b**) QTLD, (**c**) DTLD, and (**d**) DTRD.

Figure 11 shows that the damage of beech specimens with a direction perpendicular to fibers is increased with the increase of the strain rate, which can reveal the strain rate sensitivity of beech wood with a radial direction. In addition, Figure 11c,d represent the final deformation shapes of disk specimens under high strain rates of 1800 s$^{-1}$ and 2000 s$^{-1}$, respectively, which produced large deformation; some parts at the edge of the specimen were ejected and the deformed shape of the rest of the specimen resembled a rectangle. As shown in Figure 13b, the internal structure of beech specimens consists of fibers with an irregular arrangement. Under impact loading, the specimens exhibit larger deformation stretching along the direction perpendicular to fibers with the increase of the strain rate. Simultaneously, the vessel elements are compressed and then collapsed, the thickness of the specimen decreases, and the fibers fracture and then appear delaminated, which is caused by tensile stress and shear stress. As shown in Figures 11d and 13c, the fibers fracture and crack into different sizes and shapes, and are oriented in different directions. Figure 14a gives the diagram of failure mode, where the fibers are intertwined after fracturing. Therefore, the failure mechanism of the beech specimen in the radial direction can be illustrated by the decrease in the specimen thickness and deformation stretching perpendicular to fibers, together with fiber fracturing, tensile fracturing, and shear fracturing.

Figure 12 presents another failure mode, where the beech specimen with a fiber direction stretches alongside the radial direction of a specimen under impact loading, together with fiber fracturing around the specimen. As the strain rate increases, the fibers exhibit more buckling and fracturing, together with shear fracture. The damage around the specimen is worse than that in the middle position of the specimen. Figure 13a shows that the fibers close to the interface between the test specimen and incident bar buckle and break irregularly, fiber fracturing appears near to the fiber buckling, and a crack develops in the internal specimen. It can also be observed that the internal fibers fracture because of the tensile force between fibers. The beech specimen along the fiber direction under a high strain rate will be divided into pieces, as shown in Figure 14c. The failure mechanism of the beech specimen in the fiber direction can be illustrated by the decreasing thickness and deformation stretching around the specimen, together with fiber buckling, fiber fracturing, and shear fracturing, which is dependent on the strain rate sensitivity.

## 4. Conclusions

This paper studies the effect of the strain rate and beech fiber direction on the mechanical properties of beech wood under quasi-static loadings and dynamic compressive loadings. The experimental results not only reveal that the strain rate has a significant effect on the mechanical behavior of beech wood along both the LD and RD directions, but also that the mechanical properties of specimens along the LD are significantly different from those along the RD, and the elastic modulus, yield stress, and ultimate stress of specimens along the LD direction are all higher than those along the RD direction. This can be explained by their different failure modes based on the macroscopic and microscopic deformation structures. The beech wood, as a natural protective material, also shows excellent dynamic mechanical properties and energy absorption abilities at high strain rates. These findings can fill the gap in knowledge regarding the mechanical properties of beech wood, especially considering the dynamic strain rate effect, and can be applied in the material design for protective and energy dissipation structures, as well as the structure safety analysis of beech wood.

**Author Contributions:** S.P., Y.L. (Yingjing Liang), and S.H. conceived and designed experiments; W.T., Y.L. (Yijie Liu), and H.Q. performed the experiments; S.P. and W.T. analyzed the data; S.P. and Y.L. (Yingjing Liang) contributed to the original draft of this article.

**Funding:** This work was supported by the National Natural Science Foundation of China (Grant No. 11702067), the Natural Science Foundation of Guangdong Province (Grant No. 2016A030313617), and the Scientific Research Projects of Municipal Universities in Guangzhou City (Grant No. 1201620277).

**Acknowledgments:** The authors are grateful for the support from the Analyzing & Testing Center of Guangzhou University, which provided the scanning electron microscope (SEM).

**Conflicts of Interest:** The authors declare no conflict of interest.

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
