# Peer review of "Effect of the Strain Rate and Fiber Direction on the Dynamic Mechanical Properties of Beech Wood"

_forests, doi:10.3390/f10100881_

Round 1
Reviewer 1 Report
In this paper the authors investigate how strain rate affects the mechanical properties of beech wood by Split Hopkinson Pressure Bar (SHPB) Method together with conventional compression test. As a reviewer I had questions and found some issues during reviewing this paper. I believe those points should be improved prior to publish as a scientific paper. Especially, my questions should be answered in the manuscript since I believed my questions were common among the other readers of this paper. Thus, I recommend to the editors Major Revision.
Question 1
Why did the authors want to explore dynamic mechanical properties on their beech wood?
According to introduction, in ancient days beech wood in china was used for various purpose (building, ship, bridge and expensive furniture) and today it is popular for construction material. The authors should describe how strain rate effects, the main subject of this paper, are important for the utilisation of beech wood when the authors want to do for in practical purpose such as to promote usage of beech wood etc.
In introduction (L31-32) the authors write about energy absorbing material. In the description what do the authors target? I cannot see any specific usage of beech wood as an energy absorbing material, especially with the high strain rate up to 2000 S^-1.
The authors mention uses for bridge (L82-88). Do the authors focus ancient bridges? Or, in China, is wooden bridges, which need energy absorption in high strain rate (eg, bridges for automobiles), still constructed?
On the other hand, when the objectives of the authors is to explore the mechanical properties of wood, shouldn't they test much more species? Moreover, I wonder if they submit a paper with such an objective to a journal in material science or wood science.
I believe that the journal "Forests" is a journal for forestry and forest ecology, not for engineering. The authors should think who the reader of their paper is. I strongly recommend enhancing introduction by the objectives of this study in detail.
Question 2
Why did the authors take SHPB test to obtain dynamic mechanical properties rather than more common dynamic test in wood science?
In my knowledge the application of SHPB method are not so common in wood science. When the authors want to identify the mechanical properties of beech for as basic data to use beech as construction materials for buildings, they can employ impact bending test. I have no knowledge of Chinese Standards but ISO, ASTM and other standards have rules for impact bending or the other useful test methods.
I don't claim that the authors should take impact bending but do request to the authors to describe the reason of taking SHPB in their study.
If the authors just want to investigate the effects of strain rate and beech is just a material easy to obtain, the worth of this paper will be degraded because the effects of strain rate have already been investigated as seen in introduction section.
Something for the reason can be found at the end of introduction. However, current descriptions in introduction are difficult to understand or confusable. I would request to make objectives of this study clearer, and let the readers follow easier. A suggestion can be arranging order of descriptions, say, firstly give details for current usage of beech wood in Chinese wood and construction industry; secondary tell the lack of dynamic data; then describe the reason of taking SHPB test; next review current knowledge on SHPB test on wood; discuss advantage of SHPB test in the authors' objectives of study, and so on.
Question 3
The authors write in line 108-109 "the difference of mechanical properties... between radial direction and the tangential direction is not significant". Is this statement true? It may be depending on what "significant" mean but the difference in properties between the two direction is apparent in many tree species. For example, Bucur (2006, Acoustics of Wood 2nd ed, Springer) shows 137, 22.4 and 11.4 10^8 N / m^2 for longitudinal, radial and tangential Young's moduli for beech (unknown but would be European beech, Fagus sylvestris) wood of 750 kg/m3 density in Table 4.1A, representing radial is double of tangential. Many reports on various species have shown such large differences between the two direction. I don't want to claim to make experiments strictly dividing the two direction but would request to improve the description around L107-112.
Next question is "can QTRD in this study indicate the results in young's modulus from radial direction? or perpendicular to grain?" In Fig 4 the loading direction seems not to radial but diagonal between radial and tangential. Moreover, the deformation after testing shows a large effect of shear, not only compression stress. The results from this type of sample for compression test would be difficult to compare against SHPB test.
Question 4
How many specimens did the authors used in each experiment? I cannot find how many samples in the authors' experiments. The number of specimens is one of important information for the reliability of the results from an experiment. As shown in introduction, the effect of density to mechanical properties can be considered. However, there is no description in this paper on the density effect. Fig. 1 illustrates the authors seem to take their specimens from one specimen timber or log. When the density variation among specimens was small, it should be presented, eg, density variation can describe like using standard deviation around L99. Note that density is variable even within one log.
Question 5
Photographs in Fig 3 and 12 bring questions.
In Fig 12 c and d represent rectangle specimen, not disk-shaped. This contradicts the description in L116-117.
In L103-105 and Fig 1 the authors describe the specimens for QTRD and QTLD are cubic shape. However, right side photograph in Fig 3 looks cuboid. Moreover this photograph makes me doubtful to the description in L114 "along RD".
For QTRD and QTLD many testing standards requires cuboid with loading axis parallel to the long axis of specimen. Why do the authors take cubic for the sample shape?
Question 6
In my view, the description for methodology on SHPB is not enough in this paper. Again, this journal is in forestry, not in engineering.
Firstly, I questioned if QTRD and QTLD in L213, L215 and L224-225 are mistakes. Are those DTRD and DTLD?
In L214 "aLL conducted under same impact strain rate", then four strain rates are described. Very confusable.
For fomulae 1-5, the meanings of following signs are unclear: epsilon with point (t), epsilon (t), sigma T, epsilon T (t) and sigma T (t).
In L214-215 "Basing on Eq. (1)-(5) the strain rate...are obtained". Where in the equations can I find strain rate?
I suspect that:
(1) To obtain Fig. 6b, one should determine starting point (0 in Fig. 6b) for each of strain gauge on incident bar and transmission bar. Then arrange signals into like Fig. 6b. In other words, transmitted signal and reflected signal in Fig. 6a must be slid left side as much as striking signal propagate from the gauge on incident bar to the gauge on transmission bar. Shouldn't determining procedures for zero point and the arrangement the signals be described?
(2) To control strain rate, the speed of striker bar changed. Is this true? Should this also be described?
Suspecting can make nothing. I believe all the readers welcome precise description of methodology. The authors may cite one or more reference the details of methodology but still brief description must be provided.
Other issues
Lack of species name (perhaps should be in Latin name)
I cannot find any description for the species of the materials in this study. The authors write just "beech". I would ask the authors to be clear for their sample species. It is well known that different species even within a genus show different characteristics of wood. When the species is unknown, the authors should write "beech" in this paper means some species of genus Fagus. Note that trade names of wood is very prone to be named inadequately, "beech" may mean wood from other species/genera which are distant phylogenetically.
Abbreviations are not defined in the text
I believe abbreviations must be defined where it appears firstly. I cannot see the what following abbreviations mean: SHPB; PVC; MTS.
English to be improved
In some places of the manuscript I can find English usage to be improved. Below is just some examples. The authors should check the usage through the manuscript.
L9 "...orthotropic material, beech material..." and elsewhere: maybe better if "beech wood" etc.
L16 "beech fiber direction": is "beech" necessary? Convenient terminology may be "parallel to grain".
L21 "a natural protective material", L22 "protective structure": What do those mean? Revise.
L51 "sensitivity": sensitive?
L51 "full saturation" "fiber saturation": make the description clearer.
L67 "hard wood": hardwood?
L95 "natural": naturally grown? beech trees from natural forest?
L96 "characteristic of macroscopically orthotropic": is this a characteristics of "beech" wood? not for the other wood of the other species?
L176 "true strain percentage twenty-five": what does this description mean?
L326 "vascular tracheid": do the authors say here about vessel or vessel elements? Please refer any textbook of wood anatomy on the difference between tracheid and vessel element.
Reviewer 2 Report
This paper describes a practical method to evaluate the effect of stain rate on mechanical properties of beech material and reveal the difference between the effect along the longitudinal direction and radial direction. Especially it studied the dynamic mechanical properties of the beech material compare to static properties that most investigations conducted. It is recommended for publication after considering the following comments:
Page 1, Line 18. Please specify which mechanical properties are superior.
Page 1, Line 41-42. “However…” Please add 1-2 citation to support the claim.
Page 2, Line 86, Please define the medium strain rate. For instance, a range of the value.
Page 3, Line 99, how do you obtain the values of the moisture content and density of the materials? Please give a rough explanation.
Page 3, Line 95-99. Do you have the assumption that the mechanical properties of the speech material you use are evenly distributed? Then the samples are comparable.
Page 3, Line 116. Define SHPB the first time.
Page 3, Line 126. The estimate strain rate is specifically to your system. Or based on the calculation.
Page 4, Line 159. These equations are based on your hypothesis or please give the citation.
Page 5, Line 200-201. You give the interval of elastic deformation stage and plastic evolution stage, (o’a’) and (a’c’) separately. What is the interval for softening development stage, is it (c’d’)?
Page 5, Line 219-220. Please do some explanation that why Figure 6b’s relationship meets the theory therefore SHPB test is feasible.
Page 6, Line 232. Is the DTRD in “… strain curve from DTRD” is actually “OTRD”?
Page 9, Line 276. “Sensitivity” change to “sensitive”
Page 9, Line 279. “better” is a vague word. You can change to “greater” for example.
Page 9, Line 280. Why Figure 10b not use the same pattern as Figure 10a, it will make your conclusion more clear.
Page 10, Line 295 and Line 298. It has grammatical errors using “Which” here. Please rewrite these two sentence start with “Which”.
Page 11, Line 315. Please add “schematically”. Figure 15 “schematically” shows ….
Page 12, Line 323. “As shown” and “it can be observed that” are redundant, please delete one of them.
Page 12, Line 355. Change “make up” to “bridge”.
Lase but not the least. A valuable finding of the paper is that the strain rate changes the properties along the LD is superior to RD. In Page 12, Paragraph 1 and 2 (Line 321-345), you discussed the mechanism of the failure in both direction separately. However, do you have any explanation why LD is superior to RD based on the proposed mechanism? If possible, add such discussion in this part.
Reviewer 3 Report
There are many grammatical errors and language issues in the manuscript. Would need to correct those errors and improve the language, necessary for professional and accurate presentation of the research work. For example, most of the verbs in section 2.1 & 2.2 should be in past tense as opposed to present tense (verb "are" should be replaced with "were", for example), as those paragraphs describe the experiments carried out in the past. Another example of confusing and incorrect language are found in lines 95-97 (among others):
"In this study, all the test specimens are natural beech, which is the characteristic of macroscopically orthotropic materials different from isotropic materials, obtained from SHUN FENGFA WOOD Company, located in Guangdong Province, China. " To correct the grammar and to make it easy to understand, the author may consider the following edited sentence: "In this study, all the test specimens are natural beech wood obtained from SHUN FENGFA Wood company, located in Guangdong Province, China. Beech wood is characterized as a macroscopically orthotropic material rather than an isotropic material."
The motivation and contribution for this work appear to stem from the lack of high strain rate dynamic mechanical property data in the literature. Prior literature seems to have covered hard wood at medium strain rate conditions. I wonder if the author can briefly summarize the literature findings of hardwood mechanical properties (particularly, in the areas of strain rate sensitivities and failure mechanism) at medium strain rate in the introduction section. In the Results and Discussion section, I wonder if the author can further elaborate how or whether beech wood behaves differently at high strain rates versus medium strain rates. I believe that the readers would like to see clearly how the results of the work contribute to new knowledge and the significance of beech wood at high strain rates. In particular, does strain rate sensitivities at medium strain rate look similar to those at high strain rates? How is the failure mechanism at high strain rates compared to the that of medium strain rate reported in the literature?
Author Response
Point 1: “There are many grammatical errors and language issues in the manuscript. Would need to correct those errors and improve the language, necessary for professional and accurate presentation of the research work. For example, most of the verbs in section 2.1 & 2.2 should be in past tense as opposed to present tense (verb "are" should be replaced with "were", for example), as those paragraphs describe the experiments carried out in the past. Another example of confusing and incorrect language are found in lines 95-97 (among others):”
“In this study, all the test specimens are natural beech, which is the characteristic of macroscopically orthotropic materials different from isotropic materials, obtained from SHUN FENGFA WOOD Company, located in Guangdong Province, China. " To correct the grammar and to make it easy to understand, the author may consider the following edited sentence: "In this study, all the test specimens are natural beech wood obtained from SHUN FENGFA Wood company, located in Guangdong Province, China. Beech wood is characterized as a macroscopically orthotropic material rather than an isotropic material.”
Response 1: Thanks for your comment and advice. We have our manuscript checked by a professional English editing service MDPI. And we have revised the content according to your advice, details such as:
“In this study, all the test specimens were natural beech obtained from SHUN FENG FA WOOD Company, located in Guangdong Province, China. Beech wood is characterized as a macroscopically orthotropic material rather than an isotropic material. The beech specimens were machined in radial and longitudinal directions by machining. The moisture content and density of the beech material were about 10% and 0.73 g/cm3 at room temperature, respectively.”
Point 2: “The motivation and contribution for this work appear to stem from the lack of high strain rate dynamic mechanical property data in the literature. Prior literature seems to have covered hard wood at medium strain rate conditions. I wonder if the author can briefly summarize the literature findings of hardwood mechanical properties (particularly, in the areas of strain rate sensitivities and failure mechanism) at medium strain rate in the introduction section. In the Results and Discussion section, I wonder if the author can further elaborate how or whether beech wood behaves differently at high strain rates versus medium strain rates. I believe that the readers would like to see clearly how the results of the work contribute to new knowledge and the significance of beech wood at high strain rates. In particular, does strain rate sensitivities at medium strain rate look similar to those at high strain rates? How is the failure mechanism at high strain rates compared to the that of medium strain rate reported in the literature?”
Response 2: we appreciate your comment. Because there are few references related to the dynamic mechanical properties of wood, and most reference related the effect of the fiber direction, the moisture content, the temperature on the mechanical properties of wood under low strain rate. The Introduction mainly related to the summarization about the research progress of mechanics of wood under SHPB. So, we discussed the effect of strain rate and fiber direction on the dynamic mechanical properties of wood in this study. The value of minimum strain rate is 800 s-1, which still belong to low strain rate, and the value of maximum strain rate is 2000 s-1. According to the failure mechanism, we discussed the failure mechanism of beech specimens under different strain rates. The failure mechanism of wood under between high strain rate and low strain rate is different. Details can be seen in Part 3.4 Failure mechanism.

Reviewer 4 Report
This manuscript investigated the effect of strain rate and wood grain directions on mechanical properties of beech wood under quasi-static loadings and dynamic compressive loadings. To understand the effect of strain rate on the mechanical properties of beech wood, the dynamic compression tests were conducted at the strain rate range of 800 s-1 – 2000 s-1, and the quasi-static compression tests for the static mechanical properties of beech wood were also performed for a comparison. This study also investigated that the effect of grain direction on the mechanical properties by considering two loading directions, namely, perpendicular to the beech grain direction and parallel to the grain direction. The results showed that beech wood at both loading directions had significant strain rate sensitivity, and the mechanical properties of the beech wood along the grain direction were superior to that along perpendicular to the grain direction. Analysis for the macrostructures and microstructures of beech specimens was also presented the failure mechanisms. The beech wood, as a natural protective material, had special dynamic mechanical properties in the aspect of transversely isotropy. This research provides a theoretical basis for the applications of beech wood in the structural components for wood buildings.
The article is well organized and logically written. The experiments and examinations were carefully conducted. However, the manuscript can be published after the following comments be properly addressed.
There are many English grammatical errors and improper words in the manuscript, which need to be corrected. In the manuscript, “beech materials” is not a common word in the forestry industry. It is suggested to use “beech wood” to replace them. “Fiber direction” is not common. Please use “grain direction” instead. This study seems a fundamental research. It is expected to describe what the future of applications for the results and findings in this study.
Author Response
Question:
“There are many English grammatical errors and improper words in the manuscript, which need to be corrected. In the manuscript, “beech materials” is not a common word in the forestry industry. It is suggested to use “beech wood” to replace them. “Fiber direction” is not common. Please use “grain direction” instead. This study seems a fundamental research. It is expected to describe what the future of applications for the results and findings in this study.”
Response:
Thanks for your comment and advice. According to your advice, our manuscript has undergone English language editing by MDPI. And, we have revised the manuscript content, details can be seen in revised manuscript, please.
Round 2
Reviewer 1 Report
The revised version has been improved much. It would be much nicer if the authors had sent this version as the first version. Adding the background and reasons why the authors investigate beech wood by SHPB test will be very helpful for the understanding by the readers.
Despite the revision was successful, some issues including points that do not fulfil my former comments to the first version.
Thus, I would recommend that the authors check my comments again, not only current comments but also the comments to the first version, and consider the second revision.
Line numbers in the below is from "forests-564997 revised clean version.pdf".
#1 "beech"
For example in L8 and L11 the authors write "beech". Is "beech wood" better? "the mechanical properties of beech" in L11 may seem to be talking about beech trees. I believe that the authors focus on beech wood, not living beech trees. At the other places in the manuscript, there are many "beech" that should be replaced by "beech wood".
#2 beech wood as material of this study
I understand well by the authors' response and the new version of manuscript that the material beech wood was imported from Germany. If this beech is European beech, Fagus sylvatica (it should be because Germany has a lot of resources of this species and produces the wood), and the authors can confirm, it must be better to write the fact.
In addition, "natural beech" (L103) is a bit strange word usage for me. Can't it be improved like "naturally grown beech wood imported from Germany"?
#3 number of specimens
I can see the numbers of specimens in the authors' response to the question 4. However, I cannot find in the manuscript "forests-564997 revised clean version.pdf". Such information for only reviewer contributes nothing. The authors must describe these important informations for the reader of this paper.
Another question: were all the specimens taken from one log? Which position in the log was used for sample taking? It is well known that radial positions of specimens affect on various properties such as density and Young's modulus. It is also well known that different tree shows different values in wood property testing. Thus, I would recommend the authors describing above facts.
A new question arises by knowing the number of specimen. I believe each specimen show different results in static and dynamic test. Why don't the authors describe the variation between specimens in each testing condition? For example I believe the results in Fig 10 should have error bars showing the variation between species.
Note that wood is very variable material. How to present the results is the responsibility of the authors, so that I can accept Fig 10 without error bars (thinking the results with mean values only). However, I believe the author should describe how their results varies between specimens.
#4 SHPB test
By the authors' explanation in the authors' response, my understanding on SHPB test has become much better. However, important informations in the authors' response cannot be found in the manuscript. Especially, what are point epsilon(t) in formula 1, epsilon(t) in formula 2 and sigma(t) in formula 3 need to be described.
Also, the explanation in the authors' response about controlling strain rate is very informative. Why don't the authors limit such good description for reviewer only? Such information for only reviewer contributes nothing. The authors must describe these important informations for the reader of this paper.
#5 "vascular tracheid"
In wood anatomy, "vascular tracheid" refers a special type of tracheary element and can be found not so easily. I am not sure whether it occurs in Fagus. The definition by IAWA (International Association of Wood Anatomists) is:
Tracheid, vascular. — An imperforate cell resembling in form and position a small vessel member. Syn. Imperfect vessel member.
In L352, L355 and in Fig 14, the authors write "vascular tracheid". However, the cells pointed by red arrows in Fig 14b is not vascular tracheid but is vessel elements with clear perforations. I recognise that those descriptions have to be corrected.
#6 "vascular bundle"
In L33 "wood... consisting of fiber bundles and vascular bundles...".
This description is not suitable. "Wood" is secondary xylem in tree stems and does not include phloem. In other words, wood is a part of the vascular system. "Vascular bundle" refers usually a group of xylem and phloem playing transportation of water etc. in plant body. A good example of vascular bundle can be seen in transverse section of bamboo.
I don't think "fiber bundle" is a suitable expression here.
This is an issue of biology. The authors may argue they do material science but they should follow plant science when they describe phenomena in the context of plant science.
The authors write in the authors' response 6.7: "looked at other literature and used the word “vascular tracheid”. The reference in text is detailed as...". However, ref no 10 (unfortunately I could not access to the reference) seems to be on sisal and sisal is not a tree.
My suggestion is, omit "consisting of fiber bundles and vascular bundles". The phrase is not necessary.
A comment on the sample shape in static test
The authors tested again in cuboid specimens. It is OK.
I asked why the authors had their experiment with cubic specimens. I didn't want to claim they must do with cuboid specimens. Just wanted to say they should describe the reason if they did in the other way from some standards.
In addition, it would be better to describe the authors followed Chinese standards if the authors did in the second static test.
The referencing system seems not work well in the file "forests-564997 revised clean version.pdf". The citing numbers appear without brackets and without separations between two references.
